# Controlling Morphology and Physio-Chemical Properties of Stimulus-Responsive Polyurethane Foams by Altering Chemical Blowing Agent Content

**DOI:** 10.3390/polym14112288

**Published:** 2022-06-04

**Authors:** Sayyeda Marziya Hasan, Tyler Touchet, Aishwarya Jayadeep, Duncan J. Maitland

**Affiliations:** 1Shape Memory Medical Inc., Santa Clara, CA 95054, USA; 2Department of Biomedical Engineering, Texas A&M University, College Station, TX 77843, USA; tytouchet@tamu.edu (T.T.); djmaitland@tamu.edu (D.J.M.); 3Materials Science and Engineering, University of California, Berkeley, CA 94720, USA; aishwaryajai@berkeley.edu

**Keywords:** shape memory polymer foam, blowing agent, polyurethane ureas

## Abstract

Amorphous shape memory polymer foams are currently used as components in vascular occlusion medical devices such as the IMPEDE and IMPEDE-FX Embolization Plugs. Body temperature and moisture-driven actuation of the polymeric foam is necessary for vessel occlusion and the rate of expansion is a function of physio-chemical material properties. In this study, concentrations of the chemical blowing agent for the foam were altered and the resulting effects on morphology, thermal and chemical properties, and actuation rates were studied. Lower concentration of chemical blowing agent yielded foams with thick foam struts due to less bubble formation during the foaming process. Foams with thicker struts also had high tensile modulus and lower strain at break values compared to the foams made with higher blowing agent concentration. Additionally, less blowing agent resulted in foams with a lower glass transition temperature due to less urea formation during the foaming reaction. This exploratory study provides an approach to control thermo-mechanical foam properties and morphology by tuning concentrations of a foaming additive. This work aims to broaden the applications of shape memory polymer foams for medical use.

## 1. Introduction

In recent years, shape memory polymers (SMP) have shown promise in the medical device and tissue engineering space where device design primarily relies on the shape changing properties of SMPs that allows for a minimally-invasive surgical procedure [1,2,3]. SMPs are stimulus-responsive materials that undergo a shape change upon exposure to an external stimulus such as heat, light, pH, or solvent [4,5]. Thermally actuated SMPs can be mechanically programmed to have a secondary shape after the material is heated above the glass transition temperature (T_g_) and subsequently cooled below T_g_ while the mechanical strain is maintained [6]. The SMP will hold this temporary, secondary shape until it is exposed to a thermal stimulus, upon which it will return to the original, primary shape. This shape change phenomenon allows for SMPs to be programmed into small geometries and return to their expanded original shape when heat is applied.

Porous SMPs have additional benefits for their use in medical devices, such as serving as scaffolds that allow for tissue ingrowth and regeneration [7,8,9]. Dalton et al. developed hydrogel-coated SMP foams for use in vascular grafts because the porous foam helps with fluid transfer through the graft promotes tissue growth [10]. Additionally, the programmability of the SMP foam would allow for the graft to be delivered via a catheter while in a crimped shape, and a thermal stimulus would cause it to expand at the target site [10]. 

Pfau et al. developed a “self-fitting” bone scaffold with optimized mechanical properties and a controlled degradation profile to treat irregular craniomaxillofacial bone defects [11]. The SMP foam would conform to the shape of the irregular bone defect as it expands when heated using warm saline. Once the SMP foam cools down to body temperature, the scaffold would become rigid and load bearing while conforming to the irregular bone defect. Additionally, the porous structure of the implant would aid tissue growth, and native bone tissue would replace the implant as the polymer degrades over time. 

IMPEDE and IMPEDE-FX Embolization Plugs are FDA-approved devices that utilize the self-expanding properties of an SMP foam to occlude diseased blood vessels [1,12]. The low-density foam has a high capacity for volumetric filling while exerting minimal radial force upon the vessel anatomy. The porous structure of the expanded foam alters vessel hemodynamics, creating fluid stagnation that ultimately leads to stable thrombus formation and obstructed blood flow [13]. The SMP foam used in the IMPEDE product family was designed by the Maitland group and is classified as a thermoset amorphous polyurethane urea with low density and high shape recovery [14,15]. Multiple biocompatibility studies in various animal models have shown the material to have a positive tissue healing response [2,7,8]. Additionally, the SMP foams were optimized to be oxidatively degradable such that over time the polymeric plug is replaced with collagenous connective tissue [1]. The degradation byproducts of the SMP foam were evaluated by Herting et al., showing a low likelihood of a cytotoxic response in vivo [16]. 

In this work, we modify the thermo-mechanical properties of the SMP foam type used in the IMPEDE device family by altering the concentration of a foaming additive, specifically a chemical blowing agent. Water is traditionally used as a chemical blowing agent in polyurethane foam synthesis [17,18,19,20]. During the polyurethane foaming reaction, water molecules will react with free isocyanate (NCO) groups to generate carbon dioxide (CO_2_) and primary amines [19]. CO_2_ serves as a source of pore nucleation and imparts porosity onto the polymer foam while the primary amine further reacts with surrounding NCO groups to form a urea linkage [19]. Therefore, the polyurethane foams that use water as a chemical blowing agent have some degree of urea groups present within the polymer network. Additionally, it is expected that urea functionalities within the polymer backbone will increase with increasing water content due to the generation of more free amines that will be available for reaction. 

It is important to understand how material properties can be altered and controlled so that the polymer becomes better suited for its intended application. Previous works by Singhal et al. have focused on varying the chemical constituents of the polymer backbone to tune SMP foam hydrophobicity, T_g_, and actuation rate [15,21]. Similarly, Hasan et al. evaluated changes to the thermo-mechanical and morphological properties of SMP foams upon incorporating nanofillers into the polymer network [22]. However, changes to material properties due to tuning foaming aids, which are not structurally incorporated into the polymer backbone, has yet to be studied for this chemistry of SMP foam. Research by the Ryszkowska group has shown that morphology and thermal properties of viscoelastic polyurethane foams can be modified by adjusting the isocyanate index and the water concentration [23,24]. These SMP foam systems are designed specifically for medical applications such as respiratory devices and orthopedics. Therefore, in this study, we examined the outcomes of changing concentration of water, a chemical foaming aid, during SMP foam fabrication and the resulting effects on pore morphology, T_g_, mechanical properties, and actuation profiles. This work is specifically tailored towards fine-tuning SMP systems used in embolic occlusion applications. 

## 2. Materials and Methods

### 2.1. SMP Foam Fabrication

N,N,N’,N’-Tetrakis(2-hydroxypropyl)ethylenediamine (HPED, 99%; Sigma-Aldrich Inc., St. Louis, MO, USA), triethanolamine (TEA, 98%; Sigma-Aldrich Inc.), hexamethylene diisocyanate (HDI, TCI America Inc., Portland, OR, USA), Vorasurf DC 198 (Dow Corning), Vorasurf DC 5943 (Dow Corning), DABCO T-131 (Evonik Industries AG), DABCO BL-22 (Evonik Industries AG), Enovate 245fa Blowing Agent (Honeywell International, Inc., Charlotte, NC, USA) deionized (DI) water (ASTM Type II; LabChem, <1µS/cm), HPLC Grade Water (PHARMCO, 0.5 µS/cm), and isopropyl alcohol (IPA, ≥99.5%; Fisher Scientific) were used as received.

SMP foams were synthesized using the traditional gas-blowing process [25]. First, an NCO pre-polymer was synthesized with specific molar ratios HPED, TEA, and HDI. This pre-polymer was cured for 32 h at 50°C in an oven. Next, a hydroxyl (OH) pre-polymer was synthesized using the remaining molar equivalents of HPED and TEA. The final SMP foam consisted of 50 molar equivalents HPED, 50 molar equivalents TEA, and 100 molar equivalents HDI. The OH pre-polymer also contained fixed concentrations of catalysts (BL-22 and T-131) and varying molar ratios of water, a chemical blowing agent. For foam fabrication, the NCO pre-polymer was mixed with the OH pre-polymer in the presence of fixed concentrations of surfactants (DC198 and DC5943) and a physical blowing agent (Enovate 245fa). The resulting foam mixture was poured into 500 mL plastic beakers and cured for 2 min at 50 °C. The cured foam was allowed to cold-cure, under ambient conditions, for 24 h before post-processing. Table 1 shows the weight percent of each component used for foam synthesis.

Post-synthesis, the bulk foam was cut into 2 cm long cylinders with 6 mm diameter using a biopsy punch. The samples were annealed at 90 °C for 30 min and allowed to cool down completely before further processing. Foam cylinders were cleaned in 32 oz glass jars using one 30 min sonication wash in HPLC-grade water and four 30 min sonication washes in IPA. After each wash, the solvent was discarded and the jars were replenished with fresh solvent. Prior to testing, foam cylinders were dried at 100 °C under vacuum for at least 12 h, after which they were stored in a plastic storage container with dessicant. 

### 2.2. Chemical and Physical Characterization 

#### 2.2.1. Scanning Electron Microscopy (SEM)

Scanning electron microscopy (SEM) was used to assess the morphology of SMP foams. To perform this analysis, three samples of SMP foams were gold sputter-coated using a 108 Auto Sputter Coater (Cressington Scientific Instruments Ltd, Watford, England) and placed on double-sided carbon tape for imaging in a NeoScope JCM5000 (Jeol USA, Inc., Peabody, MA, USA) scanning electron microscope. Images were taken using an acceleration voltage of 10 kV in High Vacuum mode. Four different regions from each specimen were imaged to identify representative foam morphology. Pore sizes were calculated using the Image J Software (NIH, Bethesda, MD, USA). N = 25 data points were collected for axial and transverse pore size, from samples from each region, for each SMP foam formulation. 

Strut thickness of each SMP foam formulation was measured by evaluating the SEM images collected for pore size analysis. Image J software (NIH, Bethesda, MD, USA) was used to measure the width of the struts (n = 5) per formulation. 

#### 2.2.2. Fourier-Transform Infrared (FTIR) Spectroscopy

Thin foam samples were cut (2–3 mm) from bulk foams, and FTIR spectra was collected using a Bruker ALPHA Infrared Spectrometer (Bruker, Billerica, MA, USA) via the Platinum ATR Sampling Module. Sixty-four background scans of the empty chamber were taken followed by 32 sample scans of the various foam compositions. FTIR spectra was collected in absorption mode at a resolution of 4 cm^−1^ within the wavenumber range of 400 cm^−1^ to 4000 cm^−1^ with atmospheric compensation. OPUS software (Bruker, Billerica, MA, USA) was utilized to subtract the background scans from the spectra and to conduct a baseline correction for IR beam scattering and an atmospheric compensation to remove any peaks acquired due to carbon dioxide or water in the air.

#### 2.2.3. Differential Scanning Calorimetry (DSC)

Foam samples (4–5 mg) were cut from bulk foam and thermally characterized using the Q-2000 DSC (TA Instruments, Inc., New Castle, DE, USA). The first cycle consisted of decreasing the temperature to −40 °C at 30 °C·min^−1^ and holding it isothermally for 2 min. The temperature was then increased to 120 °C at 30 °C ·min^−1^ and held isothermally for 2 min. In the second cycle, the temperature was reduced to −40 °C at 30 °C·min^−1^, held isothermally for 2 min, and raised to 120 °C at 10 °C·min^−1^. T_g_ (dry) was recorded from the second cycle based on the inflection point of the thermal transition curve using TA instruments software. The aluminum pan was vented during this test to remove moisture from the sample during the first cycle. N = 3 was utilized per foam composition. 

#### 2.2.4. Thermogravimetric Analysis (TGA)

Thermal stability of the SMP foams was determined using thermogravimetric analysis (TGA). Samples (10–15 mg, n = 1) were prepared from the bulk foam. A platinum pan was used to hold each sample and tared before each run. The samples were heated from 30 to 700 °C at 10 °C·min^−1^ under nitrogen flow of 20 mL·min^−1^ using a TGA Q 50 (TA Instruments, New Castle, DE, USA). The thermograms were evaluated using TA Universal Analysis software, and percent mass remaining (%) versus temperature (°C) curves for each foam composition were graphed.

#### 2.2.5. Tensile Testing

Tensile properties of the SMP foams were determined using MTS Criterion Model 42 (MTS Systems Corporation, Inc, Eden Prairie, MN, USA), A 10N load cell was used for conducting tensile (extension) testing with a test speed of 5 mm/min. N = 5 samples were prepared for each SMP foam formulation. 2 mm thick foam slices were cut from the bulk foam using the Proxxon 37080 Hot Wire Cutter (PROXXON Inc., Hickory, NC, USA). The foam slices were further cut into dog bones using a laser engraving system (Orion Motor Tech, Lake Forest, CA, USA). The foam dog bones were endcapped with wooden blocks to prevent material damage during clamping. Strain at break (%) and Young’s Modulus (kPa) was recorded for all SMP foams with varying water content. 

#### 2.2.6. Density

Foam cubes were cut using a Proxxon 37080 Hot Wire Cutter (PROXXON Inc., Hickory, NC, USA) from top, middle, and bottom of the bulk foam as required by at ASTM standard D-3574 [26]. Length, width, and height of the cube was measured using a calibrated caliper and mass of the sample was measured using a calibrated scale. Density of each SMP foam formulation was calculated in g·cm^−3^ using Equation (1).
(1)$$Density=\frac{Mass}{\left(Length\;\times \;Width\;\times \;Height\right)}$$

### 2.3. In Vitro Shape Memory Behavior 

#### 2.3.1. Volume Recovery and Expansion

Cleaned foam cylinders (n = 3) were used to evaluate shape recovery and volume expansion of each SMP foam composition. A 203.20 μm diameter nickel-titanium (Nitinol) wire (NDC, Fremont, CA, USA) was inserted through the center of the sample along its length to serve as a stabilizer. The foam samples were radially compressed to their smallest possible diameter using a Machine Solutions crimper—306630-103 (Machine Solutions, Flagstaff, AZ) by heating the material to 100 °C, holding it isothermally for 15 min, and cooling the foams back down to program them to the crimped morphology. Initial foam diameter (5 measurements per sample) was measured and recorded for each sample using photos of the samples with a calibrated ruler and Image J software (NIH, Bethesda, MD, USA). The foams were placed in a water bath at 50 °C, removed after 20 min, and allowed to cool to room temperature. While in the heated water bath, videos of the crimped samples with a calibrated ruler were recorded to monitor time until full foam expansion was achieved. The final diameter of the samples (5 measurements per sample) was measured and recorded, using photos of the samples with a calibrated ruler and Image J software. Volume expansion was calculated using Equation (2), and volume recovery was calculated using Equation (3).
(2)$$\left(Volume\;expansion={\left(\frac{\mathrm{Recovered}\;\mathrm{diameter}}{\mathrm{Compressed}\;\mathrm{diameter}}\right)}^{2}\right)$$
(3)$$\left(Volume\;recovery={\left(\frac{\mathrm{Recovered}\;\mathrm{diameter}}{\mathrm{Original}\;\mathrm{diameter}}\right)}^{2}*100\%\right)$$

#### 2.3.2. Actuation Profiles

Cleaned foam cylinders (n = 3) were used to evaluate the actuation profile of each SMP foam composition. Samples were prepared using the method described in Section 2.3.1 and the crimped foam was actuated in a water bath at 37 °C, to be representative of body temperature. Samples were imaged every 10 s for a total of 10 min, after which they were removed from the water bath. The images were evaluated using Image J software and foam cylinder diameter was plotted against time to create an actuation profile. 

## 3. Results

### 3.1. Pore Structure and Foam Density

Figure 1 shows the variation in foam morphology as a result of adjusting water content. Less water during SMP foam synthesis yielded pores that were not inter-connected, as presented in Figure 1a,b, along with thick polymer struts due to less bubble generation and entrapment. However, as water content increased to greater than 5%, Figure 1c–f, the pores became more interconnected and membranes with pinholes appeared. Additionally, the polymer struts decreased in thickness as there were increased quantities of bubbles to fill the space within the mold. 

Table 2 shows decreasing strut diameters for SMP foams as water content increased. Increased water content also resulted in larger pores, as there was sustained CO_2_ generation from the chemical reaction between water and NCOs, which caused the pores to continue expanding while the polymer was curing into a solid foam. For example, a foam made with no water had an axial pore size of 913 ± 117 µm, while a foam made with 25% water had an axial pore size of 1670 ± 287 µm. Foam density decreased with increasing water content, Table 1, as there was less polymeric material and more void space per cubic centimeter (cc). Bubbles for the 0% water formulation were imparted only from the physical blowing agent, Enovate, which resulted in more polymeric material per cc and an overall higher density (0.169 ± 0.0016 g/cc) compared to the foams made with 25% water and Enovate as the chemical and physical blowing agents, respectively, which had the density of 0.017 ± 0.0009 g/cc.

### 3.2. Chemical and Thermal Characterization

As discussed in the introduction, increasing water content during SMP foam synthesis is hypothesized to result in increasing urea content in the polymer. Figure 2 shows absorbance spectra of the polyurethane/urea SMP foams. The amide III stretch (~1237 cm^−1^), the C-H stretch (~2919 and 2852 cm^−1^), and the N-H stretch (~3307 cm^−1^) contribute to the aliphatic polymer network consisting of HDI, HPED, and TEA. The peaks around 1689 cm^−1^ and 1646 cm^−1^ are exclusively from the contributions of urethane and urea carbonyls (C=O), respectively. Peak height of the urethane carbonyl decreased with increasing water content while a urea shoulder emerged and grew in intensity as well, with more water. These spectra confirmed that urea content in the SMP increased with higher water content used during foam synthesis. 

Similarly, Table 3 and Figure 3 show higher T_g_ values as water content increased in the polymer foam. SMP foam with no water had a T_g_ of 52 ± 1 °C, while a foam with 25% water had a T_g_ value of 64 ± 1 °C. This data confirms that as urea content in the polymer backbone increases, the T_g_ increases as well. The hydrogen bonding between urea groups is stronger compared to that between urethane groups, and therefore more energy is required to disrupt this intermolecular force in polymers with a higher urea concentration. The T_g_ results from this study confirm that there are stronger hydrogen bonding forces within SMP foams synthesized with more water, indicating the presence of more urea functional groups within the polymer backbone. 

Additionally, thermal degradation results for the SMP foam series, characterized in this study, show a consistent degradation temperature for all foams (Figure 4). Based on the inflection points of the mass loss curves, all foams degraded around 300 °C. The amount of urea groups in the polymer network did not have a major impact on the thermal degradation temperature of our SMP system. 

### 3.3. Mechanical Characterization

Table 3 shows the changes in mechanical properties, including strain at break (%) and tensile modulus (MPa), as water content increased during foam synthesis. Foams with no water had low elongation, with strain at break of 14% ± 1%, while foams with the highest water content of 25% had an elongation of 83% ± 6%. However, a decreasing trend in tensile modulus was observed as water content increased. Modulus values decreased drastically from 12.12 MPa ± 1.62 MPa, for 0% water foams, to 1.74 MPa ± 0.06 MPa, for foams with just 5% water. Tensile modulus continued to decrease to 0.14 MPa ± 0.02 MPa for foams with 25% water. 

Figure 5 shows the relative stress (MPa) versus strain (mm/mm) curves for each of the SMP formulations. SMP foam with 0% water did not tolerate a high strain compared to the foam made with 25% water. However, the peak stress for foam made with 0% water was significantly higher than that of 25% water. Additionally, the stress versus strain curves show that foams with higher water content had greater elongation compared to their counterparts made with less water.

These differences in mechanical properties are related to the strut thickness of the SMP foams as influenced by water content. Foams with thicker polymer struts, namely the ones made with less water, had low elongation and high tensile modulus, whereas the foams with thin struts displayed high elongation but low tensile modulus. Thin struts allowed the foam to tolerate a higher strain and be more flexible versus foam with thicker struts. The thick and rigid polymer struts for foams with 0% water were less susceptible to tensile deformation, which resulted in a faster break. 

### 3.4. In vitro Shape Memory Behavior

The SMP foams displayed a high volume recovery for all formulations (0% to 25% water) as shown in Table 3. Foams with 0% water had volume recovery of 106% ± 12% and volume expansion of 5× ± 1×, while foams with higher water content of 10% to 25% had volume recovery greater than 100% and volume expansion greater than 20×. This discrepancy in volume expansion is due to limited compressibility of the 0% water foams. Thicker polymer struts of the 0% water SMP formulation prevented the foam cylinder to be crimped to a small diameter that was comparable to that of the 25% water formulation. This allowed for foam cylinder to recover to its original size, however the expansion was limited as it did not undergo a significant size change. Therefore, the volume expansion of the 0% water foam was significantly smaller than the other SMP foam formulations because of its high material density. The 5% water formulation had volume recovery of 81% ± 7% and expansion of 17× ± 3×, which is lower than the other SMP foams in this series. The 5% water foam could be crimped to a small diameter cylinder, however the thicker material struts prevented full volume recovery. The 5% water foam had greater volume expansion than that of the 0% water foam because it underwent a shape change from a small, crimped diameter to a large, expanded diameter upon exposure to a heat stimulus. 

Actuation profiles of the SMP foams, in 37 °C water, were different based on their water content, Figure 6. It is important to note that foam actuation was driven by heat and not solvent. The SMP foams evaluated in this study have a similar chemical make up as the 0TM foams made by Singhal et al. and undergo a depression in T_g_ due to water plasticization of the urethane/urea network [15]. The disruption of intermolecular hydrogen bonding, by water molecules, between the urethane and urea linkages results in lower foam T_g_ in aqueous conditions, which mimics the physiological environment. Therefore, even though the dry foam T_g_, in this study, ranges from 52–65C, the wet T_g_’s become suppressed upon exposure to moisture. However, SMP foam actuation in aqueous conditions is nevertheless driven by heat from the elevated water temperature rather than the solvent itself. 

The 0% water SMP foam had the fastest recovery due to its high density, which limited the crimp diameter that could be achieved for this formulation. The 0% water foam actuated in 50 s, as indicated by the inflection point of the recovered diameter versus time curve in Figure 5. 5% and 10% water SMP foams, had similar actuation profiles with an actuation time of approximately 120 s. These foams expanded much slower compared to the 0% water foams, likely because the high material density prevented water permeation through the sample, resulting in a delayed actuation. Unlike the 0% water formulation, the 5% and 10% water foams achieved small crimp diameters of approximately 1.1 mm which, when coupled with thick foam struts, resulted in a slower actuation rate. Similarly, the 15% and 20% water SMP foams had comparable recovery profiles, with both formulations actuating at around 90 s. Notably, 15% and 20% water foams had faster actuation rates compared to 5% and 10% water foams because of smaller strut sizes, which resulted in a lower material density and faster water permeation into the polymer network. Additionally, 25% water SMP foam achieved an even faster actuation, of approximately 65 s, compared to other SMP formulations (5–20% water), since it had the smallest strut thicknesses and lowest material density. Overall, all SMP foams achieved full actuation within 5 min, or 300 s, of submersion in 37 °C water. 

## 4. Discussion

Modifying the amount of water that was used during SMP foam fabrication had a significant effect on the subsequent material properties such as density, strut thickness, T_g_, and elongation. Foams with less water typically had smaller pores with thicker struts which resulted in a high material density. Thicker foam struts further impacted mechanical properties as shown in Table 3 and Figure 4. Lower elongation and high tensile modulus were seen in SMP foams with less water content because the thick struts prevented the material from being deformed. Alternatively, foams with thinner foam struts, because of less water content in the SMP, had higher elongation, lower tensile modulus, and lower material density. When evaluated all together, these material properties played a significant role in the shape memory behavior of the foams. 

Actuation profiles of the SMP foams were primarily influenced by the material density and strut thickness rather than the material T_g_, which was the case in previous studies conducted by Singhal et al. and Hasan et al. [14,15]. Increasing water content in the SMP foam inherently increased urea quantity in the polymer network, as qualitatively seen in Figure 2. Similarly, foams with more water content and urea formation had higher T_g_ values. This modification of the polymer composition, primarily the quantity of urethane versus urea linkages, was initially hypothesized to affect shape memory behavior of the material system. However, this work showed that pore morphology and material density were much stronger contributors towards modulating actuation rates of the SMP foams. As shown in Figure 5, SMP foams with thicker foam struts and high material density had slower actuation rates compared to their low-density counterparts. Additionally, larger pore sizes of the lower density foams (15–25% water content) aided rapid actuation. This study provided us with an additional tool for controlling material properties, while not drastically modifying the polymer backbone structure, by adjusting the amount of chemical blowing agent that is used during the foaming process. 

Future works would focus on coupling the ratios chemical and physical blowing agents to achieve porous SMP systems with similar pore structure and density. Since physical blowing agents do not react with any of the foaming monomers, we can compensate for the lack of a chemical blowing agent by increasing the quantities of the physical blowing agent that is used during synthesis. Therefore, we may be able to achieve consistently low-density foams with varying T_g_ values and urea content to evaluate if increased urea impacts the shape recovery profile of our SMP foams.

In summary, we developed a series of SMP foams with tunable pore morphology and thermo-mechanical properties by modifying the amount of chemical blowing agent that was used during foam synthesis. In effect, we did not severely alter the polymer network structure, but were able to achieve controlled foam actuation profiles which were influenced predominantly by pore morphology. This work showed that we were able to alter the physical and chemical properties of SMP foams by adjusting a foaming additive that is not incorporated into the bulk polymer, whereas previous works have focused on modifying the network structure to achieve similarly tunable material properties. 

## Figures and Tables

**Figure 1 polymers-14-02288-f001:**
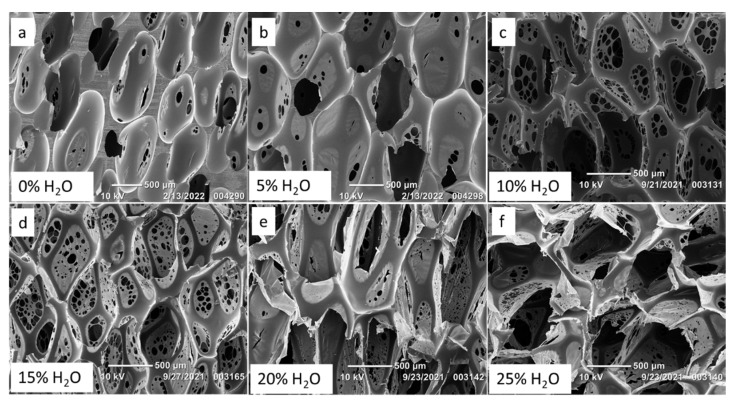
Scanning Electron Microscope (SEM) images of SMP foams synthesized with varying molar percentage of water. Increasing pore sizes and pinhole formation was observed in SMP foams (**a**–**f**) with increasing water content.

**Figure 2 polymers-14-02288-f002:**
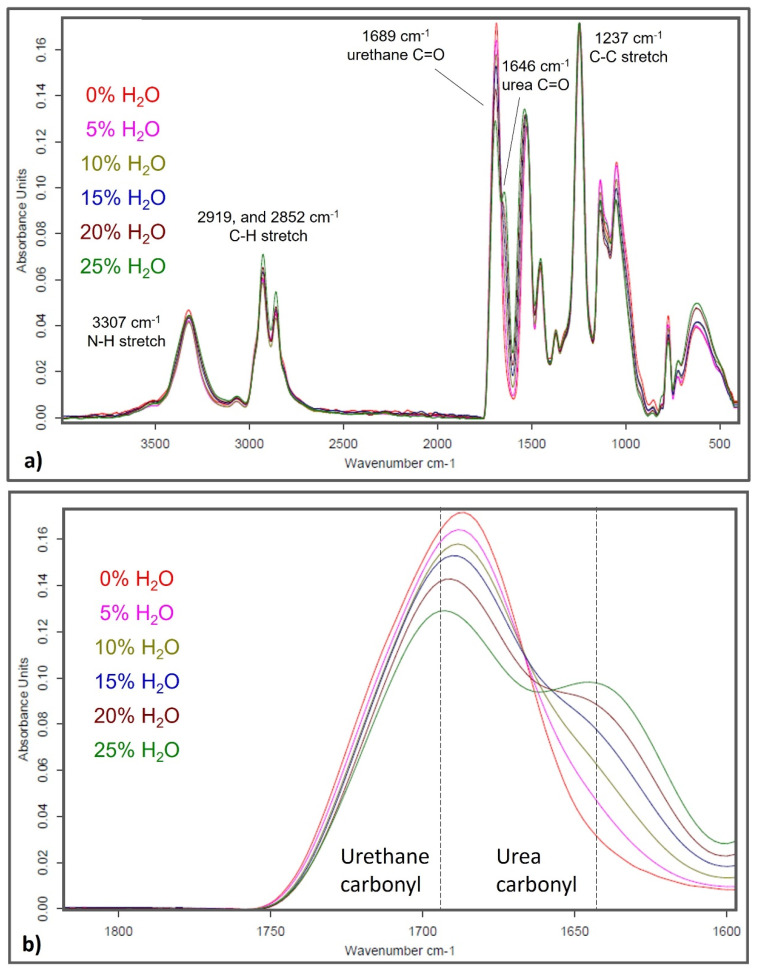
(**a**) Full absorbance spectra of SMP foams with varying water content. (**b**) Zoomed in spectra of SMP foams around 1600–1800 wavenumber (cm^−^^1^) to show varying urethane and urea peak intensities as influenced by water content.

**Figure 3 polymers-14-02288-f003:**
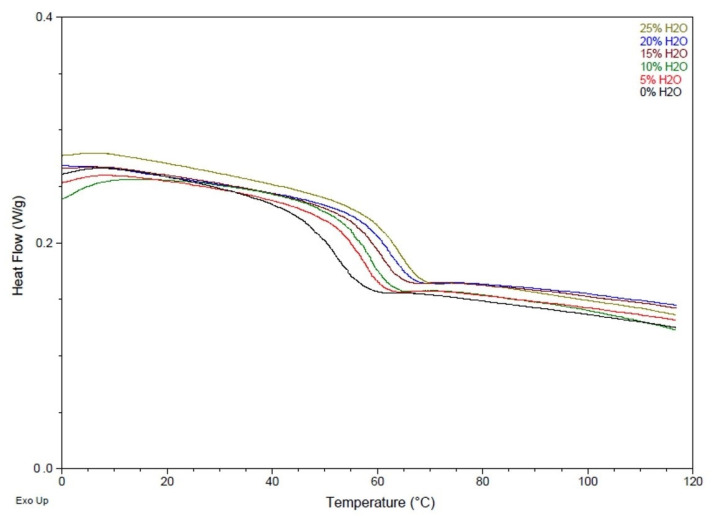
DSC thermograms of SMP foams with varying water content.

**Figure 4 polymers-14-02288-f004:**
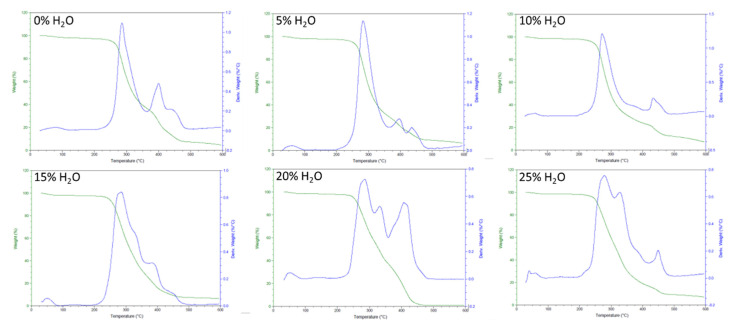
TGA thermogram of SMP foams with varying water content.

**Figure 5 polymers-14-02288-f005:**
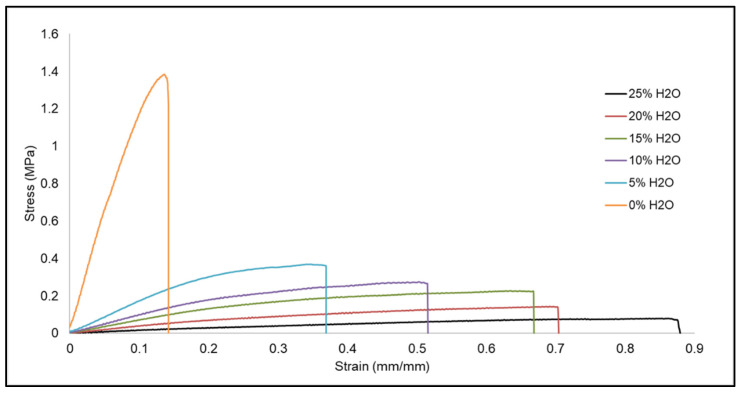
Relative stress (MPa) versus strain (mm/mm), for tensile testing, of SMP foams with varying water content.

**Figure 6 polymers-14-02288-f006:**
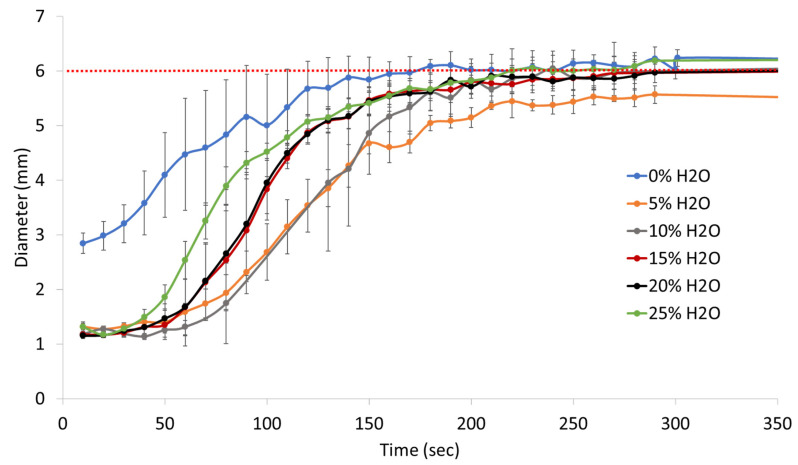
In vitro actuation profiles of SMP foams with varying water content in 37 °C water. The red dotted line indicates the original foam cylinder diameter of 6 mm.

**Table 1 polymers-14-02288-t001:** Weight percentage (%) of monomers and additives used during foam synthesis.

Water Content (mol%)	Weight Percent (%)			
	HDI	HPED	TEA	Water
25	64.97	13.83	9.41	3.47
20	62.42	16.05	10.90	2.67
15	60.11	18.01	12.25	1.94
10	57.94	19.81	13.58	1.26
5	55.88	21.64	14.74	0.59
0	53.83	23.38	15.91	0.00

**Table 2 polymers-14-02288-t002:** Pore size and density of SMP foams with increasing water content.

Water Content (mol%)	Axial Pore Size (µm)	Transverse Pore Size (µm)	Strut Thickness (µm)	Density (g/cc)
0	913 ± 117	479 ± 110	121 ± 31	0.169 ± 0.002
5	1107 ± 181	685 ± 139	101 ± 9	0.056 ± 0.001
10	1021 ± 165	659 ± 100	80 ± 26	0.035 ± 0.001
15	1188 ± 170	700 ± 105	89 ± 8	0.027 ± 0.001
20	1457 ± 214	848 ± 91	71 ± 11	0.021 ± 0.001
25	1670 ± 287	844 ± 105	62 ± 7	0.017 ± 0.001

**Table 3 polymers-14-02288-t003:** Physical properties of SMP foams with varying water content.

Water Content (mol %)	Strain at Break (%)	Tensile Modulus (MPa)	T_g_ ( °C)	Volume Recovery (%)	Volume Expansion (x)
0	14 ± 1	12.12 ± 1.62	52 ± 1	106 ± 12	5 ± 1
5	36 ± 8	1.74 ± 0.06	57 ± 1	81 ± 7	17 ± 3
10	49 ± 6	0.92 ± 0.04	59 ± 1	100 ± 6	27 ± 3
15	76 ± 13	0.66 ± 0.04	62 ± 1	102 ± 4	27 ± 3
20	73 ± 7	0.31 ± 0.06	63 ± 1	104 ± 6	28 ± 3
25	83 ± 6	0.14 ± 0.02	64 ± 1	108 ± 6	23 ± 3

## Data Availability

Raw data, supporting the results presented in this article, will be made available upon request.

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
