# Peer review of "Controlling Morphology and Physio-Chemical Properties of Stimulus-Responsive Polyurethane Foams by Altering Chemical Blowing Agent Content"

_polymers, 2022, doi:10.3390/polym14112288_

Round 1

Reviewer 1 Report

The manuscript by Hasan et al. reports on a study investigating the influence of the concentration of the chemical blowing agent water upon the physical, physio-chemical and functional properties of a polyurethane foam. It is a very nice fundamental work.

Here are my comments as suggestions for possible improvements:

  • Abstract: Better use another expression for “aqueous actuation”.
  • In terms of the foam synthesis, could you please specify the water content, above which water will no longer serve as a reactant?
  • Are there any hints in how far water content influences the open cell content?
  • Some recent work on shape memory polyurethane foams is not mentioned (exp Polym Lett 6 (2012) 63-69, Polymer Chemistry 8 (2017) 5039, Polymers 12 (2020) art. no. 1914.).
  • The different thermal stabilities of the SMP foams shown in Figure 3 are not comprehensible. Do they suggest changes in the ratio of hard to soft segments? Please comment.

Reviewer 2 Report

  1. In the literature analysis, the authors presented a modest study of the works on changes in the structure and properties of viscoelastic foams (or shape memory foams) caused by the difference in the amount of the blowing agent. Materials recently presented two papers on viscoelastic foams with different cell structures:

Okrasa, M.; Leszczyńska, M.; Sałasińska, K.; Szczepkowski, L.; Kozikowski, P.; Majchrzycka, K.;
Ryszkowska, J. Viscoelastic Polyurethane Foams for Use in Seals of Respiratory Protective Devices. Materials 2021, 14, 1600. https://doi.org/10.3390/ma14071600

Grzęda, D.; Węgrzyk, G.; Leszczyńska, M.; Szczepkowski, L.; Gloc, M.; Ryszkowska, J. Viscoelastic Polyurethane Foams for Use as Auxiliary Materials in Orthopedics. Materials 2022, 15, 133. https://doi.org/10.3390/ma15010133

  1. Moreover, many works on SMP made of viscoelastic polyurethane foams were presented, which can be used in the preparation of the introduction.
  2. In the quantitative analysis of the foam images, too few points were used for analysis. Such an analysis should perform for a minimum of one hundred points to be reliable.
  3. There are no DSC thermograms. Without them, one can have doubts about the interpretation presented by the Authors. The introduction of more urea bonds, caused by the introduction of more water to the reaction mixture increases the stiffness of the macromolecules of the foams, causing an increase in Tg. The hydrogen bonding between urea groups is stronger compared to that between urethane groups; therefore, more energy is required to disrupt this intermolecular force in polymers with a higher urea concentration.
  4. The TGA analysis results should supplement DTG curves and their interpretation.
  5. IMPEDE and IMPEDE-FX embolisation plugs use the self-expanding properties of SMP foam to seal diseased blood vessels. Therefore, to describe their properties, it is worth using the designation of properties during compression and permanent deformations after reduction, .
